# Nanofiltration Membrane Characterization and Application: Extracting Lithium in Lepidolite Leaching Solution


**DOI:** 10.3390/membranes10080178

**Published:** 2020-08-03

**Authors:** Lin Gao, Huaiyou Wang, Yue Zhang, Min Wang

**Affiliations:** 1Key Laboratory of Comprehensive and Highly Efficient Utilization of Salt Lake Resources, Qinghai Institute of Salt Lakes, Chinese Academy of Sciences, Xining 810008, China; gaolin17@mails.ucas.ac.cn (L.G.); wanghuaiyou5515@126.com (H.W.); 18503398812@163.com (Y.Z.); 2Key Laboratory of Salt Lake Resources Chemistry of Qinghai Province, Xining 810008, China; 3School of Chemical Engineering, University of Chinese Academy of Sciences, Beijing 100049, China

**Keywords:** nanofiltration, membrane, aluminum, lithium, lepidolite leaching solution

## Abstract

This study concerns the feasibility of extracting lithium and separating aluminum from lepidolite leaching solution by nanofiltration. Four commercial nanofiltration (NF) membranes (DK, DL, NF270, and Duracid NF) were chosen to investigate ion separation performance in simulated lepidolite leaching solution. Membranes were characterized according to FT-IR, hydrophobicity, zeta potential, morphology, thickness, pore size, and hydraulic permeability to reveal the effect of membrane properties on separation. NF membranes were investigated including the retention ratio of SO_4_^2−^ and Li+, the separation efficiency of Li^+^/Al^3+^, and the effect of other cations (K^+^, Na^+^, Ca^2+^) on the separation of Li^+^/Al^3+^. The results show that DK membrane displayed the appropriate permeate flux and extremely high Li^+^/Al^3+^ separation efficiency with a separation factor of 471.3 compared with other NF membranes owing to its pore size, smooth membrane surface, and appropriate zeta potential. Overall, it is found that nanofiltration has a superior separation efficiency of lithium and aluminum, which may bring deep insights and open an avenue to offer a feasible strategy to extract lithium from lepidolite leaching solution in the future.

## 1. Introduction

Lithium has been explored for wide applications in various fields, especially in rechargeable battery technologies [1,2,3,4]. As an essential metal with rapidly increasing demand on the global energy storage market, lithium exerts an crucial role in the fulfillment of energy consumption in the future [5]. Therefore, the exploration and utilization of lithium resources such as lithium ores and salt lake brines need to be focused on and strengthened. Lepidolite (ideal formula: KLi_1.5_Al_1.5_AlSi_3_O_10_F_2_) [6] is usually considered as a lithium hard-rock ores source, which possesses a lower lithium content than spodumene, but large reserves.

Many efforts have been made to extract lithium from the insoluble aluminosilicate phase of lepidolite via roasting with additives or digesting with concentrated sulfuric acid, such as the chlorination roasting method, sulfate roasting method and sulfuric acid method [7,8,9,10]. Some of these processes may have better lithium recovery than the sulfuric acid method but always with demerits such as lower purity [11] or higher energy consumption [12,13]. This is why the sulfuric acid method is still preferred commercially.

Minerals treated by concentrated sulfuric acid need to be leached with water to obtain the lithium-containing solution with high aluminum concentration, and then a complex process for removing impurity ions is required. Various forms of aluminum are presented in the aqueous solution, such as single-core aluminum and polynuclear aluminum species formed by Al^3+^, which makes the removal of aluminum more difficult. Typically, Al^3+^ are removed from lepidolite leaching solution by adding alkali to form meta-aluminate or oxalic acid to form precipitation [14]. The disadvantage is that a large amount of Li^+^ will be adsorbed or entrained by the generated amorphous colloidal precipitation, resulting in a considerable decrease in the lithium recovery ratio. Kuang has improved this process and investigated the phase equilibria in K_2_SO_4_–Al_2_(SO_4_)_3_–H_2_O ternary systems at 278.15 K with the isothermal equilibrium method, and aluminum can be removed by forming alumen with K^+^ [15], but the chemical stoichiometry of Al^3+^/K^+^ is required to be 1:1. However, the mass ratio of Al^3+^/K^+^ in the leaching solution in this experiment is 5.37/0.66, which will result in the aluminum being removed incompletely by direct crystallization along with K^+^.

Another method for removing Al^3+^ is solvent extraction, in which sulfonated kerosene is used as diluent and P204 or P507 is used as extractant to extract Al^3+^ [16,17]. The harsh operating conditions and cumbersome procedures have increased the operating cost, resulting in solvent extraction not being widely used in the industrial production. In view of the fact that the current methods of extracting lithium or removing impurity ions such as aluminum cannot achieve the flexible operation and maximum cost reduction, an effective removal of impurity ions is the key factor to reduce the cost of lithium extraction from lepidolite

Nanofiltration (NF), as an important method of separating monovalent ions and multivalent ions because of the typical pore size (1 nm) and the fixed charged groups on the membrane surface, has been not only widely applied in water treatment processes such as wastewater treatment and purification [18,19,20], but also exhibits excellent performance in the separation of lithium and magnesium in salt lake brines. Somrani [21] investigated the separation performance of NF membrane and reverse osmosis (RO) membrane, and the results revealed that NF90 membrane can extract lithium under low pressure more efficiently than XLE membranes with a 100% retention of Mg^2+^ and 15% for Li^+^. Wen [22] used the DL membrane to extract lithium from brine with borate and sulfate, and found that Donnan repulsion, dielectric repulsion, and especially steric hindrance have a considerable influence on separation performance. Reig [23] evaluated the effect of NF on the concentration and separation of Ca–Mg from RO brine, and found that Ca^2+^ and Mg^2+^ could be concentrated by NF270 membrane at about 2.5 and 3.2 times, respectively, while producing the NaCl-rich brine. In addition, it was found that the retention ratio of NF270 for SO_4_^2−^ can reach 91%. Bi [24] found that the concentration ratio of Mg^2+^/Li^+^ can be reduced from 40 to 0.9, and the recovery ratio of Li^+^ can reach 85% when using DK membranes for Li–Mg separation in salt lake brines.

This study investigated the detailed characteristics of nanofiltration membranes and the performance about extracting lithium and separating aluminum from the lepidolite leaching solution. Four commercial NF membranes (DK, DL, NF270, and Duracid NF) were used to evaluate their separation performance of ions in simulated lepidolite leaching solution, which may offer a promising method for lithium extraction from lepidolite leaching solution in the future.

## 2. Materials and Methods

### 2.1. Separation Equipment

A lab-scale nanofiltration unit device (DSP-1812W-S, Hangzhou Donan Memtec Co., Ltd., Hangzhou, China) was used for the nanofiltration experiments. The membrane is located in a radial flow circular unit with the feed entering the center of the membrane and flowing radially outward (Figure 1). Pressures and flows are interrelated and set by manual valves. If the temperature of the circulating liquid has exceeded a set value, the heat exchanger will start to decrease the temperature. Concentrate stream J_c_ that has not passed through the membrane can be recycled to the feed. Permeate stream J_p_ can be removed or recycled to the feed tank. Sampling can be done in the feed tank and from permeation flow.

### 2.2. Membrane Materials

Four commercial NF membranes including DK, DL (Suez Environnement, Paris, France), NF270 (Dow, Midland, TX, USA), and Duracid NF (Suez Environnement, Paris, France) were investigated in this study.

Membrane samples were purchased from the manufacturers. The materials of active layer and support layer, effective membrane area, and other operating parameters are shown in Table 1. The material of Duracid NF membrane is unknown, so FT-IR characterization of the film is required.

### 2.3. Filtration of Salt Solutions

In addition to a series of characterizations of four NF membranes (DK, DL, NF270, and Duracid NF), three different ion system experiments were conducted to evaluate their separation performance. The feed solution was prepared according to the composition of lepidolite leaching solution under optimized leaching conditions through sulfuric acid method. The mass fraction of the lepidolite powder and leaching solution are (mass fraction, %) Li: 1.68, 1.63; Al: 7.55, 5.37; K: 5.07, 0.66; Na: 2.29, 0.31; Ca: 1.04, 0.07; Mg, 0.26, 0.005, respectively.

In order to reveal the feasibility of nanofiltration membrane extraction of lithium in the lepidolite leaching system, this experiment introduced a monovalent anion Cl^−^ into the solution, because nanofiltration membranes have a high retention for multivalent anions, which may have great influence on the transmission of cations. The ion concentrations of the three solution systems are (1) Li^+^: 0.0471 mol/L, Cl^−^: 0.0157 mol/L, SO_4_^2^^−^: 0.0157 mol/L; (2) Li^+^: 0.0471 mol/L, Al^3+^: 0.0399 mol/L, Cl^−^: 0.167 mol/L; (3) Li^+^: 0.0471 mol/L, Al^3+^: 0.0399 mol/L, K^+^: 0.00338 mol/L, Na^+^: 0.00270 mol/L, Ca^2+^: 0.000349 mol/L, Cl^−^: 0.174 mol/L. The feed solution is composed of aluminum chloride hexahydrate, lithium chloride monohydrate, aluminum sulfate octadecahydrate, lithium sulfate monohydrate, potassium chloride, sodium chloride, and anhydrous calcium sulfate supplied by Sinopharm Chemical Reagent Co., Ltd., Shanghai, China. Deionized water (resistivity, 18.25 MΩ∙cm) is obtained by an ultrapure water machine (UPT-II-20T, Chengdu Ultrapure Technology Co., Ltd., Chengdu, China).

Filtrations were made in a total recycling mode by circulating the permeate and the retentate to the feed vessel. The device was thoroughly rinsed with feed solution to ensure that there is no residual water in the instrument, and the membrane unit was also rinsed three times with deionized water at the end of experiment. Retention experiments were conducted at constant operating temperature, pressure, and flow rate of 296.15 K, 3.4 MPa, and 3.5 LPM, respectively. Concentrate and permeate solution were obtained after equilibration of the membrane system for 10 min. Each experiment was repeated three times to improve the accuracy and error bars were added to the graphs.

### 2.4. Characterization Methods of Membranes

It should be noted that the material of Duracid NF membrane was not provided by the manufacturer and previous study. With limited information, it is difficult to compare and select the most appropriate membrane for extracting lithium from the lepidolite leaching solution. Therefore, four NF membranes need to be comprehensively characterized by FT-IR (Nicolet iS5, Thermo Fisher, Waltham, USA), hydrophobicity (JY-82, Chengde Dingsheng, Chengde, China), zeta potential (Supass, Anton Paar, Graz, Austria), microcosmic morphology (SU8010, Hitachi, Tokyo, Japan), roughness (NanoManVS, Bruker, Karlsruhe, Germany), pore size and hydraulic permeability. All membrane samples were first cut to a suitable size, and then washed three times in an ultrasonic bath of pure water for 10 min each time to prepare for the following measurements.

### 2.5. Analytical Methods

The concentrations of Li^+^, SO_4_^2−^, Al^3+^, K^+^, Na^+^, and Ca^2+^ in the feed solution and penetrate solution were measured by inductively coupled plasma-optical emission spectrometry (ICP-OES) (ICAP 6500 DUO, Thermo Fisher, Waltham, MA, USA). The organic concentration was measured by a total organic carbon (TOC) analyzer (TOC-L, Shimadzu, Kyoto, Japan). The pH of the solutions was measured using a pH meter (S210, Mettler-Toledo Instruments Co., Ltd., Shanghai, China).

### 2.6. Calculation

The separation performance of membranes were evaluated from the perspective of ion retention ratio, lithium–aluminum separation factor, and flux.

Retention ratio, R, refers to the permeability of ions, which is the main indicator for evaluating its separation ability.
(1)$$R=\left(1-\frac{C_{P}}{C_{F}}\right)\text{×}100\%$$
where C_p_ and C_f_ are the concentration of ions of the permeate and feed solution (g/L), respectively.

Separation factor, SF, means the mass ratio of Li^+^ and Al^3+^ in the permeate and feed solution.
(2)$$\mathrm{SF}=\frac{{\left(C_{{\mathrm{Li}}^{+}}{/C}_{{\mathrm{Al}}^{3+}}\right)}_{P}}{{\left(C_{{\mathrm{Li}}^{+}}{/C}_{{\mathrm{Al}}^{3+}}\right)}_{F}}$$

SF is an important parameter for directly evaluating the performance of membrane for the separation of lithium and aluminum. When SF > 1, lithium preferentially passes through the membrane as opposed to aluminum. If the nanofiltration membrane has a low retention of Li^+^, the larger the SF value, the better the separation efficiency.

Permeate flux, J, refers to the volume of permeate permeated through the effective membrane area per unit time, reflecting the ability of the composite membrane to handle a certain concentration of solution.
(3)$$J=\frac{V}{t\cdot S\cdot 3600}$$
where V is the volume of the permeate, L; S is the effective area of the diaphragm, m^2^; and t is the time taken for sampling, h.

## 3. Result and Discussions

### 3.1. Membrane Characterization

#### 3.1.1. FT-IR

The FT-IR spectroscopy of four NF membranes is almost identical, as shown in Figure 2, which means that the raw materials for preparing these membranes are almost the same. The most striking peaks in Figure 2a were shown and assigned in Table 2 [25,26,27,28]. Four NF membranes have the characteristic peaks of polyamide: Amide I band (1650 cm^−1^), Amide III band (1410 cm^−1^), Amide IV band (690, 714 cm^−1^), O=S=O symmetric stretching peak (1152 cm^−1^), and C=C phenyl group peaks (1585, 1485, 1105 cm^−1^). According to López and Fang’ s research [26,28], finding that DK, DL, and NF270 membranes have the same basic structure of polyamide layer sitting on the top of a polysulfone layer, it can be concluded that the active layers and support layers of these four NF membranes involved with Duracid NF membrane are all made of polyamide and polysulfone, respectively.

#### 3.1.2. Contact Angle

The contact angle is determined by the microstructure of the membrane surface and the hydrophilicity of the functional group of the membrane materials. It can be seen from Figure 3 that the contact angles of water on the surface of DK, DL, NF270, and Duracid NF membranes are 36.4°, 34.5°, 25.4°, and 35.9°, respectively. The smaller the contact angle, the better its hydrophilicity, which can prevent the membrane from being contaminated by other substances more effectively. This hydrophilic repulsion makes it difficult to deposit pollutants by resisting the pollution effectively, which can prolong the service life of the membrane.

#### 3.1.3. Zeta Potential

The measurements about zeta potential were performed in 1 mM KCl solution at 298.15 K using reversible ion-selective Ag/AgCl electrode, pH between 3–10 was adjusted by 0.5 M NaOH and HCl solution, and the results are shown in Figure 4. The active layers of the four membranes (DK, DL, NF270, Duracid NF) are all made of polyamide according to Figure 2, and possess fixed dissociable carboxyl and amino groups on the surface. Therefore, the change of pH can affect the dissociation of the membrane surface groups and the distribution of negative or positive charge on the surface.

The NF270 and Duracid NF membranes have more surface charges than DK and DL membranes at neutral and alkaline conditions. Zeta potential decreases with the increase of pH, and it is positive near the isoelectric point (IEP). When ζ = 0, the charge effect disappeared, and the corresponding isoelectric point of DK, DL, and NF270 is 3.49, 3.69, and 3.33, respectively. Duracid NF membrane has no isoelectric point in this pH range, and shows a relatively large surface charge compared with the other three membranes in an acid environment, which suggests that there are more amino groups and carboxyl groups on the membrane surface.

#### 3.1.4. Scanning Electronic Microscope

As shown in Figure 5a–d, the SEM images of membranes surface indicates that the dimensions of the nodules on the membrane surface are different. The thickness of Duracid NF membrane is much larger than that of the other three membranes, and the support layer structure of DK and DL membranes is more compact than that of the NF270 and Duracid NF membranes shown in Figure 5a’–d’. The nodules diameter and thickness of four NF membranes were measured directly by instrument supporting software: “Hitachi SU8000 series Scanning Electron Microscope”. The diameter of the nodule or the upper and lower boundaries of membrane can be chosen and connected into a straight line. Then, the length of line between the two points could be directly displayed, which is approximately equal to the size of membrane surface nodules and membrane thickness by averaging multiple measurements. The corresponding order is as follows: Duracid NF > DL > DK > NF270, Duracid NF > DK > DL > NF270, respectively. The image in Figure 5a’–d’ shows that the support layer structures of four membranes are basically the same, all of which are dense layered structures with sponge-like pores, and membrane pores could be approximated as free volume inside a three-dimensional network of polymer chains [29].

The difference in manufacturing processes, surface nodules, thickness, and structure will lead to the different retention characteristics of the four kinds of membranes. A larger membrane thickness of Duracid NF membrane will increase the collision probability of ionic particles and pore walls, and the distance of solvent molecules and solute ions through the membrane layer will also be extended. However, the increase of the membrane thickness can not effectively increase the amount of solute contained in the membrane, because the solute is mostly concentrated on the side of raw liquid, and the solute concentration measured by the permeate is very low. The stronger the steric hindrance, the lower the ion transmission probability and the flux; the results of the retention experiment showing that Duracid NF membrane has a higher retention and lower flux also confirm this argument.

#### 3.1.5. Atomic Force Microscope

Duracid NF.The surface of DK, DL, NF270, and Duracid NF membranes shows a typical nodular (hills and valleys) morphology in Figure 6. The nodular morphology on the surface of the DL and Duracid NF membrane is more obvious than that of the DK and NF270 membrane under the same observation scale. The higher surface roughness is consistent with the observation results of the bigger nodule diameter on the membrane surface in the SEM images (Figure 5). The roughness of the surface will affects not only the flux of membrane, but also the interaction force during the migration of particles, which will have an important impact on membrane fouling [30]. Vrijenhoek [31] has found colloidal particles will preferentially deposit in the low-lying part of the membrane, resulting in partial membrane pore blockage, and the greater the roughness, the more severe the membrane flux attenuation. In view of the presence of a large amount of Al^3+^ in the lepidolite leaching solution, there may be a large amount of colloids, so a membrane with moderate roughness should be chosen.

#### 3.1.6. Pore size and Effective Thickness

Organic molecules can be removed by a sieving mechanism, based on the small size of the membrane pore. The pore size of membranes is often characterized by the molecular weight cut-off; the molecular weight of a molecule that is retained for 90% [32]. Therefore, the effective pore size (r_p_) of membrane can be determined by establishing a quantitative relation between MW and neutral molecule radius, such as ethanol, isopropanol, n-butanol, glucose, sucrose, raffinose, and α-cyclodextrin. Using the data of these neutral molecules, the regression curve of r_s_ (organic molecule Stokes radius) and molecular weight was established, and an equation was obtained as Equation (4) [33].
(4)$$r_{s}={0.04673\mathrm{MW}}^{0.3971}$$

Figure 7 shows the regression curve of the molecular weight and retention ratio, which was obtained by the separation experiment of different neutral molecules with different molecular weights. The pore size of the membranes can be obtained by substituting the molecular weight at 90% retention into Equation (4). The molecular weight cut-off of DK, DL, NF270, and Duracid NF membranes are 292.0, 331.3, 380.6, 146.3, respectively, and the order of membrane pore size is as follows: NF270 (0.495 nm) > DL (0.468 nm) > DK (0.445 nm) > Duracid NF (0.338 nm).

#### 3.1.7. The Pure Water Permeability

L_p_ as an important parameter of membrane structure, which is only related to the temperature. To calculate the permeability of the membrane, the pure water flux was measured at different operating pressures of 1.3 to 3.4 MPa at 293.15 K. The water permeability of the membrane can be determined by the slope of the straight line drawn by the water flux and the driving force (P, MPa). The average water permeability was calculated by Equation (5) and Equation (6):(5)$$J_{V}{=L}_{P}(\Delta P-\sigma \Delta \pi )$$
where ∆P is the transmembrane pressure, σ is the reflection coefficient, and ∆π is the difference in osmotic pressure of solution and permeate stream. Furthermore, if the both sides of nanofiltration membrane are pure water, there is no osmotic pressure and Δπ should be zero, then the pure water flux can be defined by Equation (6), the results are shown in Table 3.
(6)$$J_{W}{=L}_{P}\Delta P$$

The relationship between pure water flux and operating pressure of the four NF membranes is shown in Figure 8. It can be seen that pure water permeation flux and pressure show a stable linear relationship in the pressure range of 1.3–3.4 MPa, and the order of pure water flux and L_p_ of membranes is as follows: NF270 > DL > DK > Duracid NF. The reason that the pure water flux and L_p_ of Duracid NF membrane are significantly smaller than those of the other three membranes may be owing to its larger membrane thickness and smaller pore size.

Table 4 shows the properties of four NF membranes active layers investigated in this study. The results show that these membranes have large differences in surface zeta potential, membrane thickness, surface roughness, and pore size. NF270 membrane presented the lowest intrinsic membrane resistance, being the loosest membrane evaluated in this study, while Duracid NF membrane has the thickest membrane and smallest pore size. The properties of DL and DK membranes are similar, but the structure of DK is more compact than DL.

Although a series of characterizations are carried out on DK, DL, NF270, and Duracid NF membranes, it is still hard to determine which membrane is the most suitable for lithium recovery and aluminum–lithium separation in the lepidolite leaching system in terms of the above results. Therefore, it is indispensable to evaluate the ions retention performance of four NF membranes.

### 3.2. Retention Experiments

The anion of lepidolite leaching solution by sulfuric acid method is SO_4_^2−^, which cannot pass through the nanofiltration membrane, so Cl^−^ was introduced into the solution to study the separation performance of the membranes. The results in the previous study [37] show that the concentration of Li^+^ in permeate will be reduced if SO_4_^2−^ exist. Thus, in the process of industrial operation, clear lime water and excess CaCl_2_ solution can be added into the lepidolite leaching solution to adjust the pH of the solution and completely remove SO_4_^2−^ in the solution, and the feed solution becomes pure chloride ion or with little SO_4_^2−^ solution system. Therefore, retention experiments on the three solution system using these four membranes were investigated to evaluate the retention to SO_4_^2−^, Li^+^ Al^3+^, and other cations existing in the leaching solution, and the separation efficiency of Li/Al by four NF membranes. The physical and chemical properties such as the diffusion coefficient and radii of the ions involved in the experiment are shown in Table 5 [38].

#### 3.2.1. Separation of Li^+^ and SO_4_^2−^

The retention ratio of Li^+^ and SO_4_^2−^ by four NF membranes in the presence of Cl^−^ and SO_4_^2−^ in the solution were investigated, raw liquid was prepared based on the concentration of the lepidolite leaching solution, the concentration of Li^+^ in the solution was 0.0471 mol/L, the molar ratio of Cl^−^/SO_4_^2−^ was 1:1, and the pH of the solution was measured to be 5.57. Visual MinteQ (ver. 3.0) was used to simulate ion species in the solution, which were calculated by the standard databases in the chemical equilibrium program under a temperature of 298.15 K, as shown in Table 6.

Retention experiments were conducted at constant operating temperature, pressure, and flow rate of 296.15 K, 3.4 MPa, and 3.5 LPM, respectively, and the experimental observations are shown in Figure 9. The results show that DK, DL, NF270, and Duracid NF membranes all have a high retention ratio for SO_4_^2−^, which was stable at more than 95%, but the retention ratios of Li^+^, flux, and permeate pH are quite different, as shown in Table 7.

It can be seen from Table 6 that a small part of Li^+^ has combined with Cl^−^ and SO_4_^2−^, but 96.698% of lithium still exists in the aqueous solution in the form of Li^+^. The NF membranes exhibited a high retention for SO_4_^2-^ because their active layers are made of polyamide, which possesses hydrolyzable carboxyl and amino groups, and shows a negative charge on the surface at the solution with pH = 5.57. A strong Donnan repulsion between the negatively charged membrane surface and the high-valence SO_4_^2−^ was generated, and the penetration of Li^+^ will also be affected to remain electrically neutral. The retention ratio order of Li^+^ and SO_4_^2−^ is as follows: Duracid NF > DK > DL > NF270, while the order of flux is completely reversed: NF270 > DL > DK > Duracid NF, which is consistent with the order of pore size. The high Li^+^ retention and low flux of Duracid NF membrane can be attributed to the largest membrane thickness and smallest pore size. The permeate pH of four membranes was lower than that of the raw material liquid, because H^+^ was favored to pass through the membrane with its small size and small absolute charge.

#### 3.2.2. Separation of Li^+^ and Al^3+^

In order to intuitively reveal the separation efficiency of four NF membranes on aluminum and lithium, an aluminum–lithium solution with pure Cl^−^ as anion was prepared based on the composition of lepidolite leaching solution. The concentration of Li^+^ was 0.0471 mol/L, and Al^3+^ was 0.0399 mol/L in the solution. Meanwhile, the pH of the solution was measured to be 3.33. Ion species in the solution were calculated by the standard databases in the chemical equilibrium program of Visual MinteQ (ver 3.0) under a temperature of 298.15 K, as shown in Table 8.

Retention experiments were conducted at a constant operating temperature, pressure, and flow rate of 296.15 K, 3.4 MPa, and 3.5 LPM, respectively, and the experimental observations are shown in Figure 10. The retention ratio of Li^+^ and Al^3+^, separation factor, flux, and permeate pH of four NF membranes are shown in Table 9. When the anions in the solution are all Cl^−^, the order of the four membranes to ion retention ratio was still as follows: Duracid NF > DK > DL > NF270, and the order of flux was still reversed: NF270 > DL > DK > Duracid NF, while the separation factor of aluminum and lithium is completely different: DK > Duracid NF > DL > NF270.

The retention ratio of Al^3+^ of DK, DL, and NF270 membrane is much higher than that of Li^+^, which can be attributed to the following three reasons: (1) the hydration radius of aluminum (0.475 nm, as shown in Table 5) is larger than lithium and closer or larger than the pore size of the membrane (as shown in Table 4), so the steric hindrance effect is greater for aluminum [22]. Besides, as shown in Table 8, there is single-core aluminum combined with oxhydryl or Cl^−^ such as AlOH^2+^, Al(OH)_2_^+^, Al(OH)_3_ (aq), Al(OH)^4−^, and AlCl^2+^, and polynuclear aluminum species like Al_2_(OH)_2_^4+^ and Al_3_(OH)_4_^5+^ presented in the solution, which makes the penetration of aluminum more difficult. (2) The diffusion coefficient of aluminum is much smaller. (3) Dielectric exclusion (DE) is generated by the interaction of polarized interfaces between ions and media with different dielectric constants, and exclusion energy is proportional to the square of the ionic valence [39], so a larger exclusion energy makes it more difficult for aluminum to pass through the membrane. The main reason for these variations in the retention of membranes is the diversity in the pore size of membranes.

The retention ratio to Li^+^ of DK, DL, and NF270 membranes decreased greatly, which shows that the charge exclusion effect caused by the presence of SO_4_^2−^ can increase the retention ratio of Li^+^ by nanofiltration membranes again. The retention ratio of Duracid NF membrane to Li^+^ was always maintained at above 90%, and the reason can be attributed to its pore size (0.338 nm), which is smaller than the hydration radius of Li^+^ (r_H_ = 0.382 nm), according to the separation experiments performed with different neutral molecules. The steric hindrance effect of Duracid NF membrane plays a major role in the permeation process of Li^+^, regardless of the presence or absence of SO_4_^2−^. The pH decrease value of the permeate was greater than that when there was SO_4_^2−^ in the solution, which revealed that the penetration of monovalent cations including Li^+^ and H^+^ was promoted when the anion in the solution was pure Cl^−^.

On the basis of the above separation experiment, it is found that four NF membranes have a great difference in the separation effect of aluminum and lithium. The high retention ratio of up to 90.1% of Li^+^ means that most of Li^+^ were trapped on the feed solution side and cannot separate lithium and aluminum effectively; the low flux further led to inefficient recovery. Therefore, Duracid NF membrane is not appropriate for extracting lithium from lepidolite leaching solution compared with the other three membranes. Meanwhile, the SF of DK membrane can reach 471.3, which is the largest of the four NF membranes, when the retention ratio of Li^+^ is 45.0%. In terms of excellent separation performance and moderate flux, it can be considered that the DK membrane has the best aluminum–lithium separation performance.

#### 3.2.3. Separation of Multi-Ion System

In order to further investigate the influence of other cations in lepidolite leaching solution on the separation of aluminum and lithium, an aluminum-lithium solution containing K^+^, Na^+^, and Ca^2+^ was prepared based on the composition of the lepidolite leaching solution (K^+^: 0.00338 mol/L, Na^+^: 0.00270 mol/L, Ca^2+^: 0.000349 mol/L). The concentration of Li^+^ and Al^3+^ in the solution was 0.0471 mol/L and 0.0399 mol/L, and the pH of the solution was measured to be 3.26. Ion species in the solution were calculated by the standard databases in the chemical equilibrium program of Visual MinteQ (ver 3.0) under a temperature of 298.15 K, as shown in Table 10.

Retention experiments were conducted at a constant operating temperature, pressure, and flow rate of 296.15 K, 3.4 MPa, and 3.5 LPM, respectively, and the experimental observations are shown in Figure 11 and Table 11. The results show that the retention ratio of four NF membranes to monovalent ions was significantly lower than that to divalent ions under the combined effect of charge effect and steric hindrance. DL and NF270 membranes had lower retention of Ca^2+^ owing to the larger pore size, and Duracid NF membrane exhibited a higher retention for monovalent ions because of the smaller pore size.

When K^+^, Na^+^, and Ca^2+^ were added to the solution, the retention ratios of DK, DL, NF270, and Duracid NF membrane to Li^+^ all showed a slight upward trend, rising by 3.0%, 1.0%, 1.1%, and 1.9%, respectively; the retention ratios to Al^3+^ decreased −0.6%, −1.9%, −1.3%, and −0.2%, respectively. The pH of the permeate also decreased slightly. The increase in Li^+^ retention ratio can be attributed to the competitive penetration effect of Na^+^ and K^+^, and Li^+^, Na^+^, and K^+^ have a smaller hydration radius and a larger diffusion coefficient, which will increase resistance to the penetration of Li^+^. As for the amphoteric metal aluminum, various forms of Al^3+^ were present in the aqueous solution, and the pH of solution dropped to 3.26 when other ions were added, which means that the free H^+^ in solution increased and the concentration of aluminum-combined species decreased. This is consistent with the results calculated by Visual MinteQ (ver 3.0), which shows that the proportion of Al^3+^ in the total solution increased from 97.716% to 97.721%, as shown in Table 10. Therefore, the Donnan effect and steric hindrance effect between Al^3+^ and membrane surface were weakened, which led to a reduction in Al^3+^ retention as a matter of course. It is worth noting that, when other cations exist in the solution, the DK and Duracid NF membranes with the smaller pore size had a greater increase in Li^+^ retention, and lesser decrease in Al^3+^ transmission. This phenomenon can be concluded that, even if the charge effect affected the penetration of ions to a certain extent, the main role was the steric hindrance effect on cations.

The change in Al^3+^ and Li^+^ retention will bring about the change of aluminum–lithium separation performance. In fact, the efficiency of these four NF membranes to separate aluminum and lithium was weakened to a certain extent, and the order of the separation factor of aluminum and lithium is still as follows: DK > Duracid NF > DL > NF270. In addition, the DK membrane also exhibited excellent retention performance for Ca^2+^ because of the similar pore size with the Ca^2+^ radius. The highest separation efficiency and suitable flux indicate that DK membrane presented the best performance for extracting lithium from lepidolite leaching solution.

## 4. Conclusions

In this study, the application of four commercial NF membranes (DK, DL, NF270, and Duracid NF) in the extraction of lithium from lepidolite leaching solution was investigated by a series of membrane characterization and retention experiments. The results showed that these membranes have large differences in surface zeta potential, membrane thickness, surface roughness, and pore size. Additionally, the diversity in membrane thickness and pore size mainly determines the separation performance and flux of the membrane. DK membrane exhibited the appropriate permeate flux and extremely high Li^+^/Al^3+^ separation efficiency compared with other NF membranes owing to its befitting pore size, smooth membrane surface, and appropriate zeta potential.

The separation factor of Li^+^/Al^3+^ using DK membrane can reach 471.3 and 75.4 in the pure aluminum–lithium solution and other ions presented in the solution, respectively, under the combined effects of charge, competition, and steric hindrance. These results offer a feasible strategy to extract lithium and separate aluminum from lepidolite leaching solution in the future, and demonstrate the validity of using DK NF membrane as a new environmentally-friendly and feasible method for extracting lithium from lepidolite leaching solution.

## Figures and Tables

**Figure 1 membranes-10-00178-f001:**
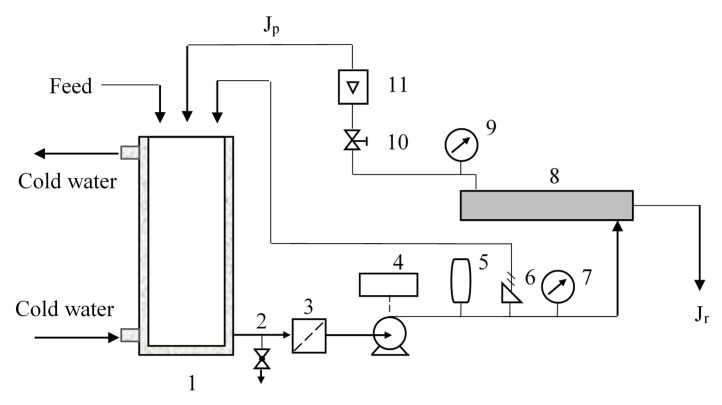
Experimental set-up of the nanofiltration separation. 1. Circulating tank; 2. Drain valve; 3. Pipeline filter; 4. Pump; 5. Frequency converter; 6. Safety relief valve; 7. Pressure gauge; 8. Membrane; 9. Pressure gauge; 10. Pressure regulating valve; 11. Concentrate flow meter.

**Figure 2 membranes-10-00178-f002:**
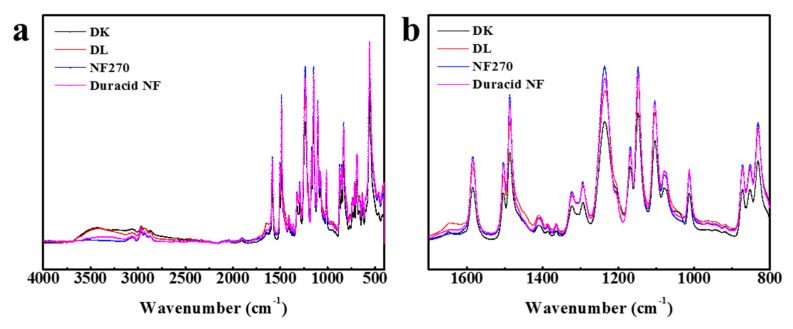
FT-IR spectroscopy for four nanofiltration (NF) membranes at wave number (**a**) from 4000 to 400 cm^−^^1^; (**b**) from 1700 to 800 cm^−1^.

**Figure 3 membranes-10-00178-f003:**
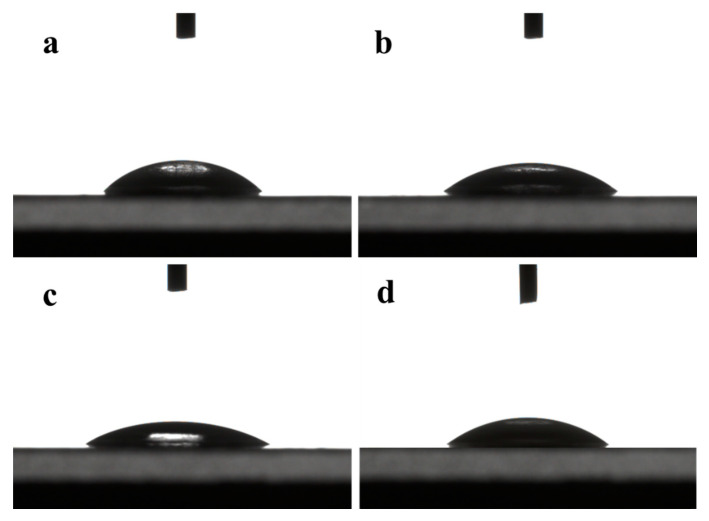
Contact angles of four NF membranes (**a**) DK; (**b**) DL; (**c**) NF270; (**d**) Duracid NF.

**Figure 4 membranes-10-00178-f004:**
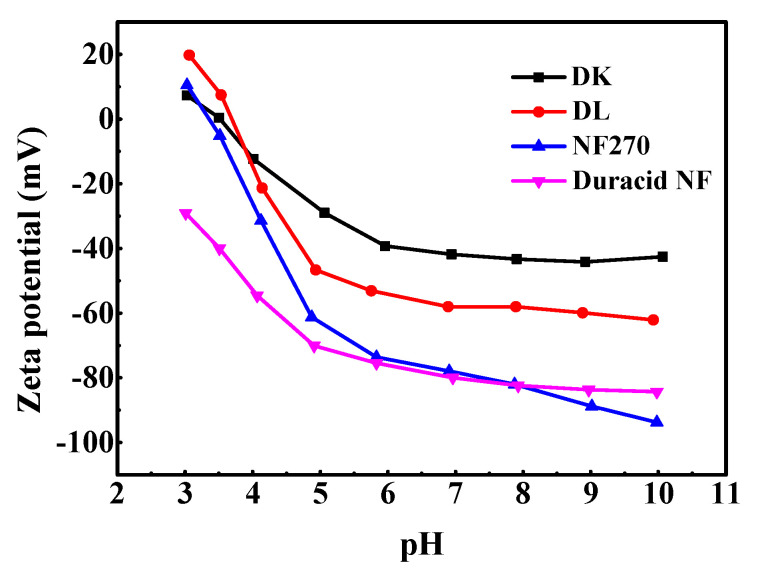
Zeta potential of four NF membranes.

**Figure 5 membranes-10-00178-f005:**
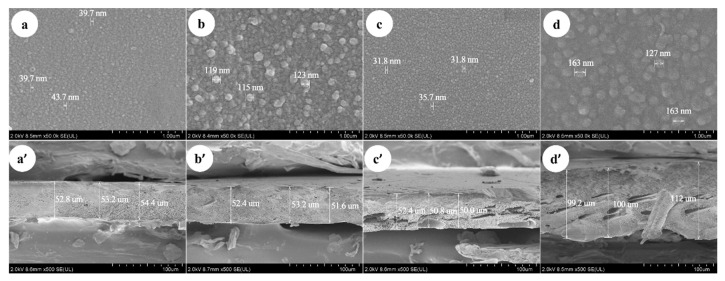
The scanning electronic microscope (SEM) of DK, DL, NF270, and Duracid NF membranes about (**a**–**d**) surface and (**a**’–**d**’) cross-sectional images.

**Figure 6 membranes-10-00178-f006:**
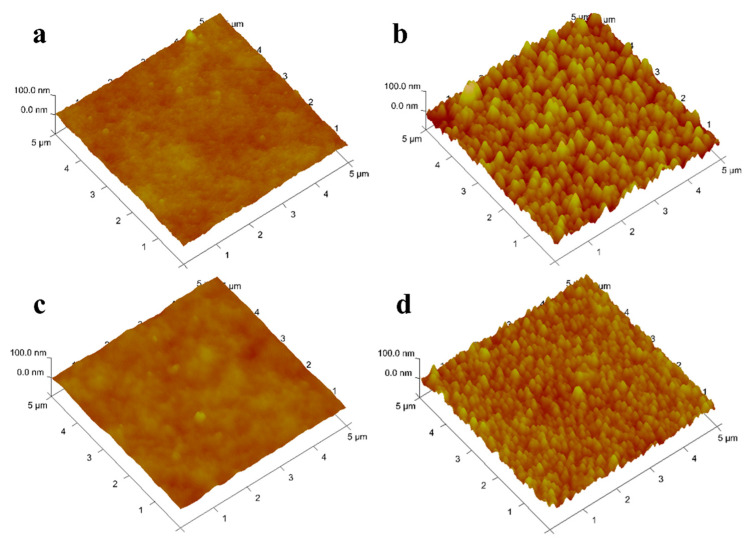
AFM images of four NF membranes (**a**) DK; (**b**) DL; (**c**) NF270; (**d**).

**Figure 7 membranes-10-00178-f007:**
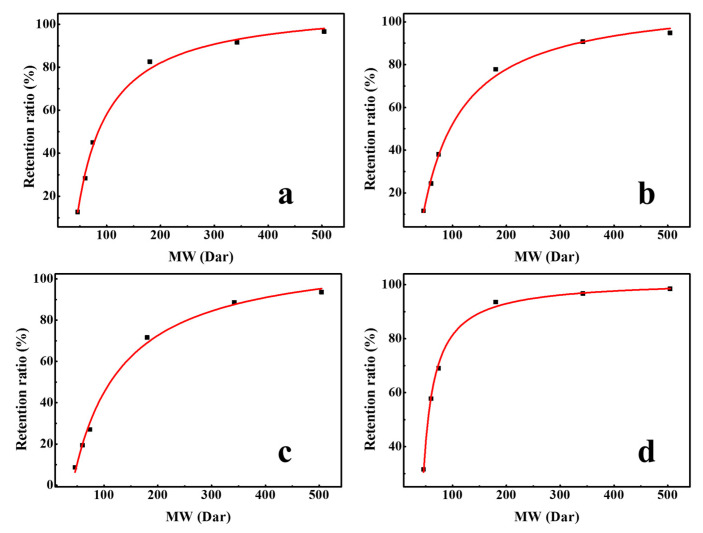
Retention ratio of different neutral molecules with different molecular weights (MWs) by four NF membranes (**a**) DK; (**b**) DL; (**c**) NF270; (**d**) Duracid NF.

**Figure 8 membranes-10-00178-f008:**
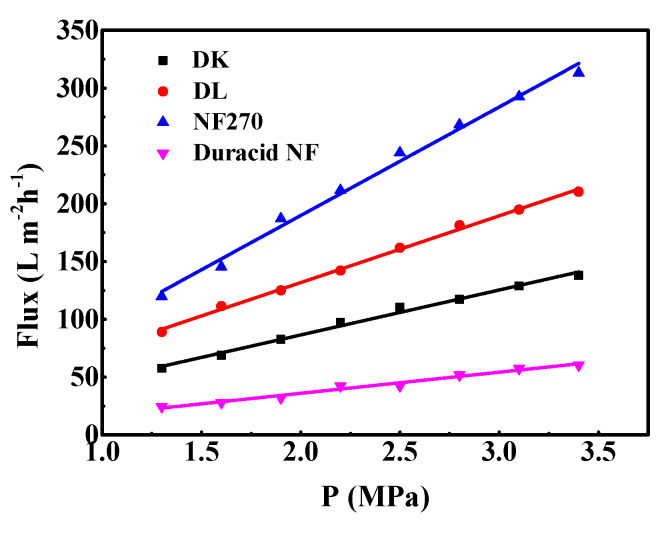
Permeate flux of pure water of four commercial NF membranes with different operation pressure.

**Figure 9 membranes-10-00178-f009:**
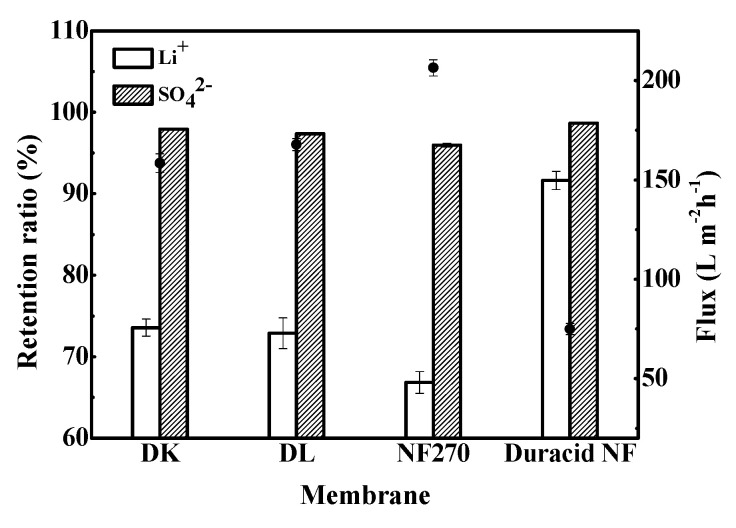
Retention ratio of Li^+^ and SO_4_^2^^−^ and flux of four NF membranes.

**Figure 10 membranes-10-00178-f010:**
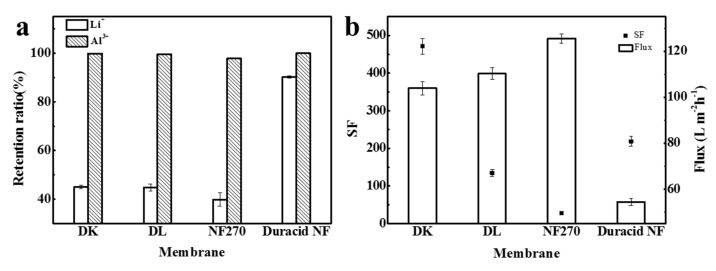
(**a**) The retention ratio of Li^+^ and Al^3+^; (**b**) separation factor (SF) and flux of four NF membranes.

**Figure 11 membranes-10-00178-f011:**
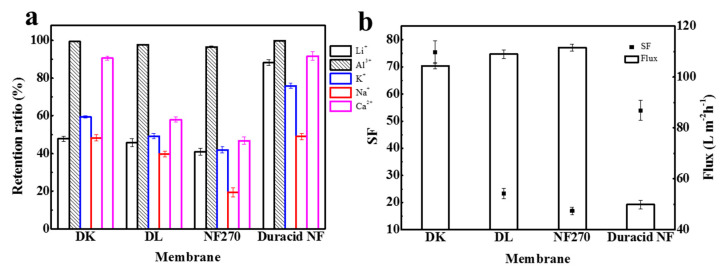
(**a**) The retention ratio of Li^+^, Al^3+^, K^+^, Na^+^ and Ca^2+^; (**b**) SF and flux of four NF membranes.

**Table 1 membranes-10-00178-t001:** The material and operation parameters of four nanofiltration (NF) membranes. PA, polyamide; PS, polysulphone.

	Active Layer	Support Layer	Pressure (MPa)	Membrane Area (m^2^)	Temperature (K)	pH
DK	PA	PS	≤ 4	0.38	≤ 323	2–11
DL	PA	PS	≤ 4	0.38	≤ 323	2–11
NF270	PA	PS	≤ 4	0.40	≤ 318	2–11
Duracid NF	-	-	≤ 8	0.38	≤ 343	< 10

**Table 2 membranes-10-00178-t002:** Peak assignment for four NF membranes [25,26,27,28].

Assignment	Wavenumber (cm^−1^)	Vibration
PA (polyamide)	2934	CH_2_ asymmetric stretching
	2864	CH_2_ attached to O or N stretching/bending
	1650	C=O stretching (Amide I band)
	1503	N–H bending
	1485	CH_2_ bending
	1410	C–N stretching coupling with NH_2_ bending(Amide III band)
	1292	CONH bending
	690; 714	N–H out-of-plane bending (Amide IV band)
PS (polysulphone)	1585	C=C Phenyl group
	1485	C=C Phenyl group
	1152	O=S=O symmetric stretching
	1105	C=C Phenyl group

**Table 3 membranes-10-00178-t003:** Hydraulic permeability of four nanofiltration (NF) membranes.

	L_p_ (m·s^−1^·Pa^−1^)		Reference
	This Study	Literatures	
DK	1.192 × 10^−11^	1.3 × 10^−11^	Straatsma [34]
DL	1.815 × 10^−11^	2.1 × 10^−11^	Bargeman [35]
NF270	2.630 × 10^−11^	4.0 × 10^−11^	Yao [36]
Duracid NF	5.012 × 10^−12^	-	-

**Table 4 membranes-10-00178-t004:** Properties of four NF membranes active layer obtained in this study.

	DK	DL	NF270	Duracid NF
Contact angle (°)	36.4	34.5	25.4	35.9
Isoelectric point	3.49	3.69	3.33	-
Thickness (μm)	53.5	52.4	51.1	103.4
Diameter of nodules (nm)	41.0	119.0	33.1	151.0
R_a_ (nm)	4.05	12.4	4.39	7.77
MWCO (Da)	292.0	331.3	380.6	146.3
r_p_ (nm)	0.445	0.468	0.495	0.338
L_p_ (m·s^−1^·Pa^−1^)	1.192 × 10^−11^	1.815 × 10^−11^	2.630 × 10^−11^	5.012 × 10^−12^

**Table 5 membranes-10-00178-t005:** Diffusion coefficients, Stokes radii, and hydrated ionic radii of ions [38].

Ions	D_s_ (10^−9^ m^2^·s^−1^)	r_s_ (nm)	r_H_ (nm)
Li^+^	1.030	0.238	0.382
Al^3+^	-	0.439	0.475
Cl^−^	2.032	1.21	0.332
Na^+^	1.333	0.183	0.358
K^+^	1.957	0.124	0.331
Ca^2+^	0.718	0.307	0.412
SO_4_^2−^	1.065	0.229	0.379

**Table 6 membranes-10-00178-t006:** Concentrations of ion species in lithium–containing solution with Cl^−^ and SO_4_^2^^−^.

Component	Concentration (mol/L)	Species Name	Concentration (mol/L)	% of Total Concentration
Lithium	0.0471	Li^+^	0.045544	96.698
		LiCl (aq)	0.000314	0.667
		LiSO_4_^−^	0.001241	2.635
Chlorine	0.0157	Cl^−^	0.015408	97.998
		LiCl (aq)	0.000314	2.002
Sulfur	0.0157	SO_4_^2−^	0.014481	92.095
		LiSO_4_^−^	0.001241	7.904

**Table 7 membranes-10-00178-t007:** Retention ratio of Li^+^ and SO_4_^2^^−^, flux, and permeate pH of four NF membranes.

	Retention Ratio (%)		Flux (L m^−2^ h^−1^)	pH of Permeate
	Li^+^	SO_4_^2−^		
DK	73.6	97.9	158.5	5.378
DL	72.9	97.4	167.8	5.436
NF270	66.8	96.0	206.4	5.231
Duracid NF	91.6	98.7	74.94	5.325

**Table 8 membranes-10-00178-t008:** Concentrations of ion species in aluminum–lithium solution with pure Cl^−^.

Component	Concentration (mol/L)	Species Name	Concentration (mol/L)	% of Total Concentration
Lithium	0.0471	Li^+^	0.044613	94.58
		LiCl (aq)	0.002557	5.42
Aluminum	0.0399	Al^3+^	0.038964	97.716
		AlOH^2+^	0.000219	0.548
		Al_3_(OH)_4_^5+^	1.9061 × 10^–5^	0.143
		Al_2_(OH)_2_^4+^	0.000113	0.565
		AlCl^2+^	0.000408	1.024
		Al(OH)_2_^+^	0.00000114	-
		Al(OH)_3_ (aq)	8.1981 × 10^–10^	-
		Al(OH)^4−^	1.5238 × 10^–12^	-
Chlorine	0.0157	Cl^−^	0.163830	98.222
		LiCl (aq)	0.002557	1.533
		AlCl^2+^	0.000408	0.245

**Table 9 membranes-10-00178-t009:** Retention ratio of Li^+^ and Al^3+^, separation factor (SF), flux, and permeate pH of four NF membranes.

	Retention Ratio (%)		SF	Flux (L m^−2^ h^−1^)	pH of Permeate
	Li^+^	Al^3+^			
DK	45.0	99.9	471. 3	103.8	3.077
DL	44.8	99.6	135.0	110.2	3.192
NF270	39.8	97.8	27.8	115.1	2.728
Duracid NF	90.1	99.9	218.6	54.4	3.015

**Table 10 membranes-10-00178-t010:** Concentrations of ion species in aluminum–lithium solution containing K^+^, Na^+^, and Ca^2+^.

Component	Concentration (mol/L)	Species Name	Concentration (mol/L)	% of Total Concentration
Lithium	0.0471	Li^+^	0.04462	94.596
		LiCl (aq)	0.0025488	5.404
Aluminum	0.0399	Al^3+^	0.038966	97.721
		AlOH^2+^	0.000218	0.548
		Al_3_(OH)_4_^5+^	0.000019091	0.144
		Al_2_(OH)_2_^4+^	0.00011262	0.565
		AlCl^2+^	0.00040686	1.020
		Al(OH)_2_^+^	1.1377 × 10^–6^	-
		Al(OH)_3_ (aq)	8.1765 × 10^–10^	-
		Al(OH)_4_^−^	1.5225 × 10^–12^	-
Potassium	0.00338	K^+^	0.0032418	96.026
		KCl (aq)	0.00013415	3.974
Sodium	0.00270	Na^+^	0.0025896	96.026
		NaCl (aq)	0.00010716	3.974
Calcium	0.000349	Ca^2+^	0.00031204	89.332
		CaCl^+^	0.000037263	10.668
		CaOH^+^	6.4718 × 10^–14^	-
Chlorine	0.0157	Cl^−^	0.16356	98.061
		NaCl (aq)	0.00010716	0.064
		AlCl^2+^	0.00040686	0.244
		LiCl (aq)	0.0025488	1.528
		CaCl^+^	0.000037263	0.022
		KCl (aq)	0.00013415	0.08

**Table 11 membranes-10-00178-t011:** Retention ratio of Li^+^, Al^3+^, K^+^, Na^+^, and Ca^2+^; SF; flux and permeate pH of four NF membranes.

	Retention Ratio (%)					SF	Flux (L m^−2^ h^−1^)	pH of Permeate
	Li^+^	Al^3+^	K^+^	Na^+^	Ca^2+^			
DK	47.7	99.3	59.4	48.3	90.6	75.4	104.5	3.054
DL	45.7	97.7	49.2	39.8	58.0	23.3	109.1	3.176
NF270	40.9	96.5	41.9	19.4	46.8	16.9	111.6	2.720
Duracid NF	88.2	99.8	75.8	49.0	91.7	53.9	49.9	3.003

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
