# Peer review of "Nanofiltration Membrane Characterization and Application: Extracting Lithium in Lepidolite Leaching Solution"

_membranes, 2020, doi:10.3390/membranes10080178_

Round 1

Reviewer 1 Report

Nanofiltration Membrane Characterization and Application:  Extracting Lithium in Lepidolite Leaching Solution

The work describes the separation factor of monovalent ions (Li(I), K(I)) from multivalent ions (Ca(II), Mg(II), Al(III)) from Li rich solutions using a group of different NF. The work is inside the journal is not have huge innovation from the membrane aspects and the innovation is associated to the application or de study of Li rich solutions. In general, de methodological procedures are reproducing sound described approaches and the main weakness are associated to the results and discussion section were are a list of things that need some clarification:

  • An effort should be done to prepare a table describing the properties of the membrane active layer, please do not use “select” layer. It is no used in membrane science for describing composite membranes
  • I do not agree all four have the same chemistry, three of them are similar (semi aromatic/aromatic polyamides), by Duracid has a sulphamide chemistry. For example, review the papers of Lopez et alt on NF membranes.
  • A more focussed review on the state of the art for NF applied to brines should be provided. As an example Reig et at, when separated or removed Ca,Mg and SO4 from NaCl is not reported.
  • In this direction the FTIR analysis should be reviewed and the sulphamide group should be identified.
  • It is not clear how the active layer thickness has been calculated and a clearer description should be provided
  •  It is not performing any speciation analysis of the brines evaluated and authors consider them as ions however most of them are complex forming species where the charge could be different. A speciation analysis is requested.
  • Of critical effect is aluminium, it is claimed to be as Al+3, and complexed to form complex with OH-, however it is also forming complexes with sulphate and chloride ions. A speciation analysis should be provided
  • The title of the paper is not describing what is occurring, Li(I) is distributed between both streams permeate and retentate?. What is the objective of this study, it is not clear?
  • A discussion on the technical implications of the separation factors should be provided? What will be the selected membrane for the study objective?
  • In some of the experiments pH was close to 3, and then protons in the same order of magnitude than other ions, I wonder if the transport if H+ was evaluated? Did author follow the variation of pH on the permeate and retentate sides.
  • Table 11 and 12 data shown are provided with two decimal units, do you think that both decimals have statistical significance?
  • Similar question is postulated to discussion on the text of ions rejection ratio, permeate flux or separation factors with two decimal units.
  • Please do not use retention rate and use retention ratio.

Author Response

Many thanks for reviewer’s comments and helpful suggestion on this manuscript. Additionally, please see the point-by-point comments in the attachment for detailed response.

Reviewer 2 Report

The comments are included in the attached Word file.

Author Response

(The authors gave the same response as above.)

Round 2

Reviewer 1 Report

Authors have covered 100% of my requests. Then I recommend the work for publication in its present form.

Author Response

Thank you very much for giving me such a high evaluation. And thanks again for what you have done.

Reviewer 2 Report

The manuscript can be accepted for publication.

Author Response

Thank you very much for giving me such a high appraisal. And thanks again for what you have done.